# Probing Genome-Scale Model Reveals Metabolic Capability and Essential Nutrients for Growth of Probiotic *Limosilactobacillus reuteri* KUB-AC5

**DOI:** 10.3390/biology11020294

**Published:** 2022-02-11

**Authors:** Thanawat Namrak, Nachon Raethong, Theeraphol Jatuponwiphat, Sunee Nitisinprasert, Wanwipa Vongsangnak, Massalin Nakphaichit

**Affiliations:** 1Specialized Research Unit: Prebiotics and Probiotics for Health, Department of Biotechnology, Faculty of Agro-Industry, Kasetsart University, Bangkok 10900, Thailand; thanawat.n@ku.th (T.N.); sunee.n@ku.ac.th (S.N.); 2Institute of Nutrition, Mahidol University, Nakhon Pathom 73170, Thailand; nachon.rae@mahidol.ac.th; 3Department of Zoology, Faculty of Science, Kasetsart University, Bangkok 10900, Thailand; 4Interdisciplinary Graduate Program in Bioscience, Faculty of Science, Kasetsart University, Bangkok 10900, Thailand; theeraphol.ja@ku.th; 5Omics Center for Agriculture, Bioresources, Food, and Health, Kasetsart University (OmiKU), Bangkok 10900, Thailand

**Keywords:** *Limosilactobacillus reuteri*, probiotic strain, genome-scale metabolic model, metabolic capability, nutrients

## Abstract

**Simple Summary:**

Considerable attention has been given to the species *Limosilactobacillus reuteri* regarding its probiotic potential. Here, the genome-scale metabolic model (GSMM) of *L. reuteri* KUB-AC5, namely *i*TN656 was developed and exploited to evaluate the metabolic capability of *L. reuteri* KUB-AC5 under various carbon sources. The simulated growth behaviors of *i*TN656 under single and double omission analysis promisingly identified 14 essential single nutrients and 2 essential double nutrients (pairwise glutamine-glutamate and asparagine-aspartate) for *L. reuteri* KUB-AC5 growth. Moreover, the integrated transcriptomic analysis using *i*TN656 allowed to probe the potential metabolic routes for enhancing growth of *L. reuteri* KUB-AC5 involved in sucrose uptake, nucleotide biosynthesis, urea cycle, and glutamine transporter. Overall, *i*TN656 provides a powerful systems biology platform for gaining insights into cell metabolism of lactic acid bacteria and guiding the rational development of synthetic media design for scale-up of probiotic production in industrial scale.

**Abstract:**

*Limosilactobacillus reuteri* KUB-AC5 displays the hallmark features of probiotic properties for food and feed industries. Optimization of cultivation condition for the industrial production is important to reach cell concentration and cost reduction. Considering the strain-specific growth physiology, metabolic capability, and essential nutrients of *L. reuteri* KUB-AC5, the genome-scale metabolic model (GSMM) of *L. reuteri* KUB-AC5 was developed. Hereby, the GSMM of *i*TN656 was successfully constructed which contained 656 genes, 831 metabolites, and 953 metabolic reactions. The *i*TN656 model could show a metabolic capability under various carbon sources and guide potentially 14 essential single nutrients (e.g., vitamin B complex and amino acids) and 2 essential double nutrients (pairwise glutamine-glutamate and asparagine-aspartate) for *L. reuteri* KUB-AC5 growth through single and double omission analysis. Promisingly, the *i*TN656 model was further integrated with transcriptome data suggesting that putative metabolic routes as preferable paths e.g., sucrose uptake, nucleotide biosynthesis, urea cycle, and glutamine transporter for *L. reuteri* KUB-AC5 growth. The developed GSMM offers a powerful tool for multi-level omics analysis, enabling probiotic strain optimization for biomass overproduction on an industrial scale.

## 1. Introduction

*Limosilactobacillus reuteri* KUB-AC5, previously known as *Lactobacillus reuteri* KUB-AC5 [1], belongs to lactic acid bacteria. *L. reuteri* is one of the predominant species serving for functional foods and feed supplements [2]. The strain KUB-AC5 was originally isolated from chicken intestine displayed the hallmark features of probiotics [3]. This strain is able to produce antimicrobial substances which inhibited both of Gram-positive and Gram-negative bacteria, such as *Escherichia coli*, *Salmonella* Typhimurium, *Salmonella* Enteritidis, *Bacillus cereus*, *Bacillus*
*subtilis*, *Micrococcus luteus*, and *Staphylococcus aureus* [3,4]. According to in vivo assay, strain KUB-AC5 could inhibit *Salmonella* infection in chicken and mice model [5,6]. In addition, this strain additionally attenuated inflammation in both gut and spleen of mice model [6]. According to health benefit properties, *L. reuteri* KUB-AC5 is therefore potentially useful for the development of new probiotic ingredient in the fields of food and feed stuffs at industrial scale.

The successful rate of probiotic therapy always relied on living cell concentration. To realize health benefits, probiotic bacteria must be viable and available in high cell concentration with more than 10^6^ colony forming units (CFU) per gram of foods [7,8]. As lactobacilli are fastidious microorganisms that have complex nutritional requirements for growth—including carbohydrates, amino acids, vitamins, nucleotides, and fatty acids—thus, optimal media formulation is the important factor for probiotic production in industry [9,10]. The most common medium for lactic acid bacteria is the DeMan-Rogosa-Sharpe medium, MRS [11]. However, MRS for industrial production would be high cost with unauthorized ingredients. The optimal design of media composition is hence important with respective to cost effective process in industrial biotechnology.

Over the past decades, technological advancements in next-generation DNA sequencing [12] have greatly explored. Previously, a genome sequencing of *L. reuteri* KUB-AC5 showed its metabolic potential associated with probiotics properties, such as the ability to synthesize cobalamin (vitamin B12) and folate (vitamin B9) [13]. Moreover, transcriptome profiling revealed gene expression of *L. reuteri* KUB-AC5 in responses to carbon sources and growth stages [14]. Beyond genomics and transcriptomics, it is interesting to move forward on revealing metabolism and its benefactions to the biomass production and metabolite formation. So far, the genome-scale metabolic models (GSMMs) are useful tools in metabolic engineering and systems biology that could help to comprehend the metabolism and physiology of the organisms. GSMMs integrate genome sequences with other high throughput omics data at systematic level which is a scaffold to connect overall metabolisms. The GSMMs of several species from *Lactobacillaceae*, such as *Lactiplantibacillus plantarum* WCFS1 [15], *Lactococcus lactis* MG1363 [16], *Limosilactobacillus reuteri* JCM 1112 [17], had been reconstructed and applied for different applications—e.g., probiotic cell factories for industrial foods and feeds [18].

In this study, a high quality GSMM of probiotic *L. reuteri* KUB-AC5 was therefore developed for enhancing growth and biomass production. To explore its overall metabolic capability, initially a metabolic network and model of *L. reuteri* KUB-AC5 was constructed and afterwards simulated the growth physiology under different carbon sources. To the end, the essential nutritional requirement was identified by integrative GSMM of probiotic *L. reuteri* KUB-AC5 with the transcriptome data. This developed GSMM can be used as a systems biology tool for gaining insights into cell metabolism of lactic acid bacteria. This study serves for guiding a theoretical foundation for designing cultivation media for further scale-up of probiotic production.

## 2. Materials and Methods

### 2.1. Metabolic Network Construction of L. reuteri KUB-AC5

To construct the metabolic network of *L. reuteri* KUB-AC5, a template-based method was used. The metabolic network of *L. reuteri* JCM1112 [17] was selected as a template in this study since high genome similarity between *L. reuteri* KUB-AC5 and *L. reuteri* JCM1112 [13] was observed. Initially, the KUB-AC5 bacterial genome sequence was retrieved from NCBI database (accession no. SRR10059212) [13]. The pairwise orthologous genes were then performed between two strains KUB-AC5 and JCM 1112 using the best bidirectional (BBH) hits from BLASTP with the following parameters—i.e., bit scores ≥ 100 and E-values of 1 × 10^-10^ as cut-offs [13,19]. Non-gene-associated reactions, including spontaneous reactions and literature-based metabolic reactions, were also manually curated and added to the model. Transport and exchange reactions were then added or deleted throughout the network connectivity. The gap-filling of metabolic reaction was performed by MeGaFiller [20]. The resulting metabolic network was further manually curated using the RAVEN toolbox 2.0 [21] and added metabolic reaction from an earlier metabolic network of *L. reuteri* ATCC PTA 6475 (*i*HL622) [18]. Additionally, the KEGG [22] and MetaCyc [23] databases were also used to improve the metabolic network based on orthologous genes. The metabolite names and reversibility of metabolic reactions were then manually curated gene-protein-reaction (GPR) associations, gene IDs, and metabolite-reaction abbreviations using KEGG [22], ChEBI [24], and MetaNetX databases [25].

### 2.2. Model Development of L. reuteri KUB-AC5

The constructed metabolic network of *L. reuteri* KUB-AC5 was subsequently converted to be a stoichiometric model. The biomass reaction was added into the model which was adopted from the template-based model of *L. reuteri* JCM1112 [17]. The contents of protein, lipid, and carbohydrate were retrieved from the earlier study [17]. The ratios of amino acids and nucleotide compositions in the *L. reuteri* KUB-AC5 biomass were estimated from its genome sequence. Vitamins and lipid composition was obtained from the other literature supports [15,26]. For the energetic parameters, the ATP requirements for the non-growth associated maintenance (NGAM) was set at 0.37 mmol ATP gDCW^−1^ [15]. The ATP cost of growth-associated maintenance (GAM) was estimated to be 49.7 mmol ATP gDCW^−1^ by fitting model simulation with experimental data for cellular growth on glucose as a carbon source.

### 2.3. Cultivation of L. reuteri KUB-AC5

*L. reuteri* KUB-AC5 obtained from the collection of the Department of Biotechnology, Faculty of Agro-industry, Kasetsart University, Thailand [4]. For inoculum preparation, the cell culture was propagated twice in MRS medium (Difco). For cultivation, the KUB-AC5 containing 10^7^ CFU mL^−1^ were suspended in 0.85% NaCl solution with the absorbance 600 nm of 0.5. The medium used for the cultivation was the modified MRS which composed of 5.0 gram per liter (g L^−1^) yeast extract, 2.0 g L^−1^ of K_2_HPO_4_, 1.0 g L^−1^ of (NH_4_)_2_SO_4_, 0.1 g L^−1^ of MgSO_4_·7H_2_O, 0.05 g L^−1^ of MnSO_4_·4H_2_O, and 1.0 mL of Tween 80 [27]. The medium was adjusted for pH 6.5 ± 0.2 before autoclaved at 121 °C under 15 pound per square inch (psi) for 15 minutes (min). The individual sterile carbon source solution (i.e., glucose, sucrose, maltose, or lactose) was added to a final concentration of 20 g L^−1^. The medium was inoculated at 2% (*v*/*v*). The cell cultivation was performed in an individual 500 mL laboratory bottle at 37  °C under static conditions. All batch cultivations were independently conducted in three biological replicates. Cell cultures were collected at 0, 3, 6, 9, 12, 18, and 24 hour (h). The dry weight measurement was obtained by pipetting 10 mL of cell culture samples filtered through 0.22 micron (µm) nitrocellulose membrane filter (Whatman laboratory Products Ltd., Maidstone, England). The residual solids were washed with 20 mL of deionized water. The filter and solids were replaced in the oven at 60 °C. After drying, the filters were allowed to cool at room temperature in a desiccator and then constant weight was achieved and further calculated for the biomass. Cells were removed by centrifugation at 12,000 revolutions per minute (rpm) at 4 °C for 4 min to obtain cell-free supernatant (CFS). The CFS was used to measure the concentrations of residual sugars. The concentration of residual sugar was determined by HPLC (Shimadzu, Tokyo, Japan) on a 4.6 × 250 mm Shodex Asahipak NH2P-50 4E column (Showa Denko K.K., Tokyo, Japan) under RI detector. The column was operated at 30 °C with acetonitrile:water ratio (75:30) using a flow rate of 1.0 mL min^−1^. The composition of the biological macromolecules of *L. reuteri* KUB-AC5—i.e., moisture content, protein, carbohydrate, lipid, and ash—were analyzed by proximate analysis followed by AOAC official method. All physiological data obtained from this study were statistically analyzed using SPSS statistical program (International Business Machines Corp. (IBM), New York, NY, USA). The average values were reported in the form of mean value ± standard error (mean ± SD) and the statistically significant difference between values were compared by Duncan’s Multiple Range Test under *p*-value < 0.05.

### 2.4. Model Validation for L. reuteri KUB-AC5 Growth Using Flux Balance Analysis 

Flux balance analysis (FBA) is a mathematical approach that is widely used for investigating optimal flux distribution through a metabolic model towards the objective function [28]. Here, the constraint-based flux simulation was performed using FBA under RAVEN 2.0, MATLAB (R2018b) using Gurobi optimizer (Gurobi Optimization Inc., Houston, TX, USA) as a linear programming solver. To estimate the optimal flux distribution under the maximized cell growth, the biomass reaction was set as the objective function. For the given carbon sources—including glucose, sucrose, maltose, or lactose—the *in silico* growth simulation was run by constraining each carbon uptake rate which was taken from our experiments throughout this study. Besides, the uptake rates of all amino acids and lipids were set as 5 mmol gDCW^−1^ h^−1^, the uptake rate of vitamins and oxygen were constrained to 1 × 10^-4^ and 1 × 10^-10^ mmol gDCW^−1^ h^−1^, respectively. The prediction results were validated with our experimental data taken from *L. reuteri* KUB-AC5 cultivation (Section 2.3.). The scripts used for model development towards validation of *L. reuteri* KUB-AC5 can be found at a public GitHub repository (https://github.com/sysbiomics/Limosilactobacillus_reuteri_KUB_AC5-GSMM, accessed on 10 January 2022).

### 2.5. Using GSMM of L. reuteri KUB-AC5 as a Scaffold for Growth Simulation and Integrative Transcriptomics Analysis

The developed GSMM of *L. reuteri* KUB-AC5 was used as a scaffold for two studies which included growth simulation for identifying the essential/preferable nutrients for *L. reuteri* KUB-AC5 growth and integrative transcriptomics analysis for identifying key metabolic routes for enhancing *L. reuteri* KUB-AC5 growth.

#### 2.5.1. Identifying the Essential/Preferable Nutrients for *L. reuteri* KUB-AC5 Growth

Enabling growth and biomass production of *L. reuteri* KUB-AC5 typically relied on undefined combinations of numerous nutrients with the excess concentrations of amino acids and vitamins. To identify the specific essential/preferable nutrients for growing *L. reuteri* KUB-AC5, the model was set to assess growth capability under the omission of 25 possible nutrients including the 20 amino acids (i.e., alanine, serine, leucine, isoleucine, phenylalanine, valine, tyrosine, histidine, methionine, tryptophan, cysteine, proline, threonine, arginine, lysine, glycine, asparagine, aspartate, glutamine, and glutamate) and the 5 vitamins (i.e., B1 (thiamin), B3 (nicotinate), B5 (pantothenate), B6 (pyridoxamine), and B7 (biotin)). The model simulation for single omission analysis was run by decreasing the uptake rate of individual nutrient out of 25 possible nutrients by a factor 0.5 at a time. For all simulations, the uptake rates of glucose, lipid and oxygen were limited to 25, 1 and 1 × 10^-10^ mmol gDCW^−1^ h^−1^, respectively. To further identify the other essential nutrients, the double omission analysis by omitted the two nutrients at a time was also performed. If the predicted infeasible growth behavior was observed in the single and double omission analysis of all 25 nutrients, then those nutrients were identified to be as essential ones. Alternately, the predicted growth behaviors of *L. reuteri* KUB-AC5 under both single and double omissions of all 25 nutrients were observed under growth rate < 0.005 h^−1^, those nutrients were considered to be the preferable nutrients for *L. reuteri* KUB-AC5 growth. In addition, identifying the favorable carbon sources of *L. reuteri* KUB-AC5 was also explored by setting the uptake rate of any 15 different carbon sources at 0, 1.0, 2.0, 4.0, 8.0, 16.0, 20.0, and 24.0 mmol gDCW^−1^ h^−1^ at a time. During the model simulations, the uptake rates of amino acid, vitamin, lipid, and oxygen were constrained to be 1, 1, 1, and 1 × 10^-10^ mmol gDCW^−1^ h^−1^, respectively. If the predicted feasible growth behavior was observed with fast growth rate, then those carbon sources were considered to be preferable ones.

#### 2.5.2. Identifying Key Metabolic Routes for Enhancing *L. reuteri* KUB-AC5 Growth

For identifying key metabolic routes for enhancing *L. reuteri* KUB-AC5 growth, the developed GSMM was probed with integrative transcriptome analysis. The list of differentially expressed genes (DEGs) under different carbon sources (e.g., sucrose and glucose) at logarithmic phase under threshold of FPKM value ≥ 1 and |log_2_ fold change| ≥ 1.5 and the false discovery rate (FDR) < 0.05 was retrieved from Jatuponwiphat et al., 2021 [14]. Then, DEGs were integrated with the developed GSMM using R package Piano (Platform for Integrated Analysis of Omics data) [29].

## 3. Results and Discussion

### 3.1. Characteristics of the L. reuteri KUB-AC5 Model (iTN656) and Comparative Analysis with other Related Models

*L. reuteri* KUB-AC5 metabolic model called *i*TN656 was developed from its genome sequence underlying the incorporation of strain-specific physiological data. This *i*TN656 model contained 656 genes, 831 metabolites, and 953 metabolic reactions amongst 2 cellular compartments—i.e., cytoplasm and extracellular space (Table 1). Overall stoichiometry of metabolic precursors involving in biomass synthesis reaction were quantitatively estimated based on macromolecular contents of carbohydrate, protein, lipid, nucleotide, and vitamin obtained from our experimental results and literature supports (Figure 1a). Regarding on genes, there were 29.9% out of the 2196 total genes from annotated data retrieved from genomic data of *L. reuteri* KUB-AC5 [13] which encoded the protein functions in cellular metabolism and membrane transport as shown in Appendix A. In comparison with the other *L. reuteri* strains in context of genomic characteristics, KUB-AC5 shared 535 and 611 common orthologous genes with JCM1112 and ATCC PTA 6475 genomes, respectively [17,18], accounted for 94.1% of total genes identified in the KUB-AC5 metabolic network. The remaining 39 genes (5.9%) were assigned as unique genes identified in the KUB-AC5 metabolic network (Appendix A). Interestingly, we observed that *i*TN656 showed that largest number of genes, metabolites, metabolic reactions, and GPRs (810 associations) in the model when compared to LbReuteri and *i*HL622 models (Table 1). Moreover, the *i*TN656 included high copy numbers of genes involved in amino acid transport systems, such as histidine transporter (13 genes), glutamine transporter (12 genes), and arginine transporter (11 genes) which suggest that KUB-AC5 can assimilate high concentration of amino acids as nitrogen and energy sources.

Of 656 genes with metabolic functions of *L. reuteri* KUB-AC5, we further categorized these functions using KEGG database [22]. The overall metabolism was divided into nine metabolic functional categories as shown in Figure 1b.

As shown in Figure 1b, we observed that the largest metabolic functional category was carbohydrate metabolism (124 genes), accounting for 18.9% of the total genes. Considering the GPR associations, these 124 genes could be assigned protein functions in the terms of the Enzyme Commission (EC) numbers, which were classified into 13 sub-categories, namely glycolysis/gluconeogenesis, pentose phosphate pathway, pyruvate metabolism, starch and sucrose metabolism, galactose metabolism, amino sugar and nucleotide sugar metabolism, butanoate metabolism, fructose and mannose metabolism, propanoate metabolism, citrate cycle (TCA cycle), pentose and glucuronate interconversions, glyoxylate and dicarboxylate metabolism, and inositol phosphate metabolism (Appendix A). These suggest that *L. reuteri* KUB-AC5 could use wide range of carbon sources [13]. It can explain how *L. reuteri* KUB-AC5 adapts to the microhabitats of the gastrointestinal environment which may enhance its survival, competitiveness, and persistence [13,30]. In comparing these three GSMMs of *L. reuteri* (Table 1) [17,18], such as in context of EC numbers, observably 355 EC numbers were commonly identified between them and 126 EC numbers were uniquely identified in *i*TN656 model (Appendix A). Interestingly, these unique enzymes are related to probiotic properties. Of these 126 EC numbers, they were majorly involved in metabolisms of cofactors and vitamins—e.g., folate biosynthesis, riboflavin metabolism, thiamine metabolism, nicotinate and nicotinamide metabolism, pantothenate and CoA biosynthesis, porphyrin metabolism, and vitamin B6 metabolism (Appendix A). These results are consistent with earlier study [13]. The production of vitamins from strain KUB-AC5 has resulted in health benefits to the host which was proposed in relation to its probiotic properties. Moreover, *i*TN656 model shows to have unique choloylglycine hydrolase (EC: 3.5.1.24) or called bile salt hydrolase protein (BSH) which is participated in de-conjugation of bile acids. According to physiological report [3,4], the strain KUB-AC5 could tolerance to high bile concentrations. Moreover, *L. reuteri* KUB-AC5 genome data contained a unique levansucrase (EC: 2.4.1.10) which was related to levan production [14]. The levan showed antimicrobial activity against pathogenic bacteria including Salmonellae, Listeriae, Pseudomonads, and certain strains of *Escherichia coli*, *Staphylococcus aureus*, *B. cereus*, and *B. subtilis* [31]. Taken together, the *i*TN656 model could guide important characteristics for screening of probiotic strains. A highlight metabolic landscape of *L. reuteri* KUB-AC5 for growth is illustrated in Figure 2.

### 3.2. Phenotypic Characteristics of L. reuteri KUB-AC5 for in Silico Model Validation

To validate the *i*TN656 model, the in silico growth rate was simulated by maximizing the biomass under four different carbon sources. In this study, glucose, sucrose, lactose, and maltose were selected for GSMM validation regarding to their implications in food and feed industries. Besides, we would like to use C6 (e.g., glucose) and C12 (e.g., sucrose, lactose, and maltose) for overall metabolic capability studies. Typically, glucose is a basal carbon source used in MRS medium which is a selective culture medium designed to favor the growth of Lactobacilli in the laboratory. For lower production costs, sucrose, which is the most abundant disaccharide, has been attracting much attention as a promising carbon source for large scale production of probiotics. In addition, lactose and maltose are the two common food disaccharides, frequently derived from milks (e.g., human milk oligosaccharides (HMOs)) and starch, respectively. The simulation data constraining the uptake rates of individual carbon source was varied from 0.940–6.634 mmol gDCW^−1^ h^−1^ (Table 2). Accordingly, the validation results of *i*TN656 model are shown in Figure 3. The results show that growth rate of KUB-AC5 in experiment and simulation agreed well when using glucose as carbon source with only 0.59% error. Moreover, the percentage of errors in maltose, lactose, or sucrose were in ranges of 0.91–9.98%. This suggests that the *i*TN656 model could be used to predict cell growth using glucose, sucrose lactose, or maltose as carbon sources. As observed, the sucrose showed to be the best carbon source for growth when compared between *in silico* and *in vitro* data (Figure 3).

Considering *L. reuteri* KUB-AC5 growth kinetic characteristics, carbon sources utilization of *L. reuteri* KUB-AC5 was performed including glucose, sucrose, lactose, or maltose (Table 2). Observably, *L. reuteri* KUB-AC5 grown under sucrose showed to gain the highest biomass production (1.650 ± 0.030 gDCW L^−1^). These experimental results clearly demonstrate that sucrose was the most preferable on the KUB-AC5 growth. Considering growth rate prediction on sucrose using model simulation, *i*TN656 suggests that the sucrose agreed well to be the most preferable carbon source which would metabolize through starch and sucrose metabolism by sucrose phosphorylase (EC: 2.4.1.7). This catalyzed by the reversible phosphorolysis of sucrose with inorganic phosphate, producing glucose 1-phosphate and fructose [32]. These metabolites can be readily converted to an intermediate in the glycolytic pathway. Interestingly, no ATP requirement in this process, phosphorolysis can be regarded as an energy-saving cellular process [33]. This specific utilization pathway resulted in a decreased ATP cost which may lead fast growth. Since sucrose phosphorylase also presents in *L. reuteri* strains LTH5448, TMW1.106, and 100–23, thereby the sucrose becomes often use for supporting the growth of these bacteria [34]. Promisingly, sucrose could be used as an alternative low-cost carbon source which is attractive for the production of whole-cell mass product with commercial effectiveness.

### 3.3. Identifying the Essential Nutrients for L. reuteri KUB-AC5 Growth through Single and Double Omission Analysis

Stimulating growth of lactic acid bacteria including *L. reuteri* was relied on the complex, semi-defined nutrients including vitamins and amino acids. Nevertheless, the specific preferable nutrients for KUB-AC5 growth remain unclear. Therefore, we used *i*TN656 to investigate how the individual nutrients and their variations contributing to *L. reuteri* KUB-AC5 growth through single omission analysis. Interestingly, this analysis enabled the identification of the 14 potential nutrients that could be essential for growth of *L. reuteri* KUB-AC5 as shown in Figure 4a. These included the four vitamins (i.e., B3 (nicotinate), B5 (pantothenate), B6 (pyridoxamine), and B7 (biotin)) and the 10 amino acids (i.e., alanine, serine, leucine, isoleucine, phenylalanine, valine, tyrosine, histidine, methionine, and tryptophan). In contrast, there remained 11 non-essential nutrients for growth which were vitamin B1 (thiamin), cysteine, proline, threonine, arginine, lysine, glycine, asparagine, aspartate, glutamine, and glutamate.

Once, the double omission analysis of nutrients was further explored. Out of 11 non-essential nutrients, more interestingly, pairwise amino acids e.g., glutamate-glutamine as well as asparagine-aspartate showed to become essential for growth. Figure 4b clearly shows that *L. reuteri* KUB-AC5 can overcome growth deficiency in the absence of glutamate or aspartate by the addition of glutamine or asparagine. Suggesting by *i*TN656 model, these amino acids—i.e., glutamine and asparagine—were the alternative substrates for synthesizing glutamate and aspartate by the activities of glutamine synthetase (EC: 6.3.1.2) and asparagine synthetase (EC: 6.3.5.4), respectively. Lactobacilli strains primarily used amino acids to fulfill their demand for growth [35] and stimulation [36]. Altogether, the results were in agreement with the previous study reported by Santos et al., 2009 [37].

Focusing on other non-essential nutrient—e.g., vitamin B1 (thamin)—the *i*TN656 model and simulation results revealed that *L. reuteri* KUB-AC5 can synthesize thiamin by thiamine salvage pathway through the catalytic activities of hydroxymethylpyrimidine kinase (EC: 2.7.1.49), hydroxyethylthiazole kinase (EC: 2.7.1.50), phosphomethylpyrimidine kinase (EC: 2.7.4.7), thiamine-phosphate diphosphorylase (EC: 2.5.1.3), and alkaline phosphatase (EC: 3.1.3.1; 3.1.3.2; 3.1.3.100). For the other amino acids—e.g., cysteine, proline, threonine, arginine, lysine, and glycine—the *i*TN656 model hereby suggested that *L. reuteri* KUB-AC5 hold a metabolic capability of producing these amino acids. For example, cysteine could be synthesized from serine and hydrogen sulfide through the conversion of serine O-acetyltransferase (EC: 2.3.1.30) and cysteine synthase (EC: 2.5.1.47; 2.5.1.65) and proline could be synthesized through the reduction of pyrroline 5-carboxylate by pyrroline-5-carboxylate reductase (EC: 1.5.1.2). In addition, glycine, lysine, arginine, and threonine could be synthesized through aspartate metabolism (Figure 4a).

Here, we further compared the in silico growth behaviors of *L. reuteri* KUB-AC5 under the omission analysis of these identified 14 essential single nutrients against the literature. As observed, the in vitro omission experiments performed by Santos et al., 2009 [37], showed that omitted leucine, phenylalanine, valine, tyrosine, histidine, methionine, or tryptophan from the chemically defined medium (CDM) at a time profoundly inhibited the growth of *L. reuteri*. This is in agreement with the simulation results of *i*TN656 suggesting that the model can be used for identifying the preferable nutrients for optimizing growth of *L. reuteri*. In addition, the metabolic capability of producing vitamin B1 (thiamin) by *L. reuteri* predicted by the *i*TN656 model was found consistent with earlier study by Saulnier et al. 2011 [38], which identified a complete pathway for thiamine biosynthesis in *L. reuteri* genome. This supporting information hereby suggests that the *i*TN656 model can facilitate the optimization of probiotic production in the fields of food and feed stuffs at an industrial scale.

### 3.4. Alternation of Carbon Sources for Optimizing L. reuteri KUB-AC5 Growth and Biomass Production

The alternation of carbon sources is crucial for optimizing *L. reuteri* KUB-AC5 growth and biomass production. Currently, the identification of preferable carbon sources that can promote the growth of probiotics has attracted more and more attention in functional food and feed industries. To explore this, the *i*TN656 model was set to assess the metabolic behavior of *L. reuteri* KUB-AC5 when grew under the different types of carbon sources, including monosaccharides (e.g., arabinose, glucose, galactose, mannitol, and fructose), disaccharides (e.g., maltose, sucrose, lactose, trehalose and melibiose), oligosaccharide (e.g., raffinose). The simulation results demonstrated that *L. reuteri* KUB-AC5 was capable of utilizing a variety of carbohydrates as carbon sources for stimulating growth. Of which, raffinose, which was known as a prebiotic oligosaccharide, was found to be the most favorable carbon source for optimizing *L. reuteri* KUB-AC5 growth as shown in Figure 5. This is consistent with the fact that raffinose is a family of non-digestible oligosaccharides that are frequently found in a large variety of seeds and legumes, such as soybean which is a major source of macronutrients in poultry feeds [39]. The simulated metabolic flux analysis demonstrated that this oligosaccharide could be hydrolyzed by raffinose galactohydrolase (EC: 3.2.1.22), yielding sucrose and galactose. Interestingly, sucrose was found to be an alternative favorable carbon source that could support higher rate of growth compared with other disaccharide sugars such as maltose, trehalose, and lactose as seen in Figure 5. Moreover, galactose and glucose were found to be more effective in promoting the growth of *L. reuteri* KUB-AC5 than using the other monosaccharides as a sole carbon source. Besides those favorable carbon sources, the *i*TN656 model demonstrated that *L. reuteri* KUB-AC5 can grow well under fructose utilization. We further validated the physiological result and found that the concentration of fructose had been decreased from 20 g L^−1^ to 11.74 ± 0.78 after 24 h by *L. reuteri* KUB-AC5. Additionally, pH of CFS was slightly decreased from 6.54 ± 0.08 to 5.84 ± 0.04. Our in vitro observation clearly proved that fructose is an alternative carbon source, even though an earlier report showed no growth according to API 50 CHL method.

In case of others, *i*TN656 model contains genes-encoding enzymes involved in the transport of glycerol (AC5u0009GL001930), mannitol (AC5u0009GL001848, AC5u0009GL000165, AC5u0009GL000668, AC5u0009GL001849 or AC5u0009GL002128), and trehalose (AC5u0009GL000409, AC5u0009GL001181, AC5u0009GL001184, or AC5u0009GL002112). Moreover, it contains enzymes involved in glycerol, mannitol, arabinose, and trehalose utilizations, such as glycerol dehydratase (EC: 4.2.1.30), mannitol dehydrogenase (EC: 1.1.1.67), arabinose isomerase (EC: 5.3.1.4), and trehalose phosphorylase (EC: 2.4.1.64), respectively. Thus, the *i*TN656 model can be used to guide *L. reuteri* KUB-AC5 growth capability.

### 3.5. Putative Metabolic Routes for Enhancing Growth of L. reuteri KUB-AC5 

As in silico and in vitro growth rates observed under sucrose utilization were found to be favorable for *L. reuteri* KUB-AC5, compared to the other disaccharides and monosaccharides. Therefore, we performed the integrative transcriptome analysis using the *i*TN656 model as a scaffold. Considering these reporter metabolites in context of expression patterns, we clearly observed the higher transcriptional changes in a form of FPKM values occurred in sucrose than glucose as illustrated in Figure 6. Interestingly, the presence of sucrose in the culture medium induced the upregulation of genes encoding sucrose transporter and sucrose phosphorylase (EC: 2.4.1.7). Moreover, the reduction of fructose to mannitol catalyzed by mannitol dehydrogenase (EC: 1.1.1.67) was selectively upregulated upon sucrose supply. This reaction was suggested to be an alternative pathway instead of lactate formation to regenerate nicotinamide adenine dinucleotide (NAD+) [40]. In addition to the sugar uptake, several metabolic pathways related to biomass precursor production, including nucleotide biosynthesis, urea cycle, and the uptake transporters of glutamine were highly expressed. Interestingly, the expression of glutamine transporter could supply more glutamate formation via carbamoyl-phosphate synthase (EC: 6.3.5.5), which the product carbamoyl phosphate is an intermediate in the biosynthesis of nucleotides through orotate pathway [41]. This implied that glutamate might be a key role for *L. reuteri* KUB-AC5 growth [14]. For example, glutamate was found to be the preferential precursor for proline biosynthesis through the catalytic activities of glutamate 5-kinase (EC: 2.7.2.11), glutamate-5-semialdehyde dehydrogenase (EC: 1.2.1.41), and pyrroline-5-carboxylate reductase (EC: 1.5.1.2). We observed that the genes encoding these reactions were also upregulated when sucrose was used as a sole carbon source. Moreover, the conversion of pyruvate to lactate by lactate dehydrogenase (EC: 1.1.1.27) was selectively downregulated upon sucrose supply, while the conversion of N(omega)-(L-Arginino)succinate to fumarate via urea cycle by argininosuccinate lyase (EC: 4.3.2.1) was simultaneously upregulated. The integrative transcriptome analysis clearly suggests that sucrose predisposes *L. reuteri* KUB-AC5 for high biomass production.

## 4. Conclusions

We presented *i*TN656, the first GSMM of *L. reuteri* KUB-AC5, and validated its growth performance with the experiments under different carbon sources. The simulated flux distributions under alternative substrate utilizations of *i*TN656 highlighted that the utilization of sucrose as a sole carbon source is the most favorable for supporting growth capability of *L. reuteri* KUB-AC5. Besides, *i*TN656 model suggests essential/preferable nutrients for *L. reuteri* KUB-AC5 growth. With integrative transcriptome analysis, the *i*TN656 model also demonstrates that putative metabolic routes—e.g., sucrose uptake, nucleotide biosynthesis, urea cycle, and the glutamine transporter—as preferable paths for *L. reuteri* KUB-AC5 growth. Our results not only demonstrated the high quality of *i*TN656, but also could support the rational design of media composition for metabolic engineering purposes in the field of the probiotic applications in industrial foods and feeds.

## Figures and Tables

**Figure 1 biology-11-00294-f001:**
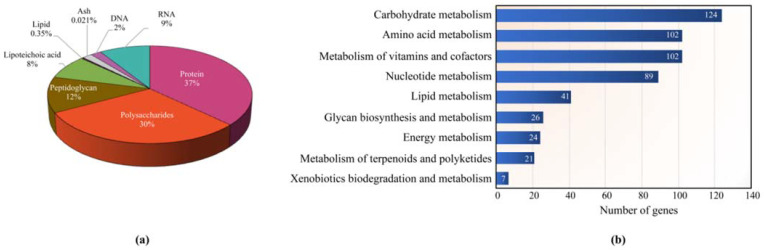
The *i*TN656 metabolic features in context of biomass composition and metabolic functional categories. (**a**) Metabolic precursors involving in biomass synthesis. (**b**) The horizontal bar chart shows the number of genes devoted to different metabolic functional categories.

**Figure 2 biology-11-00294-f002:**
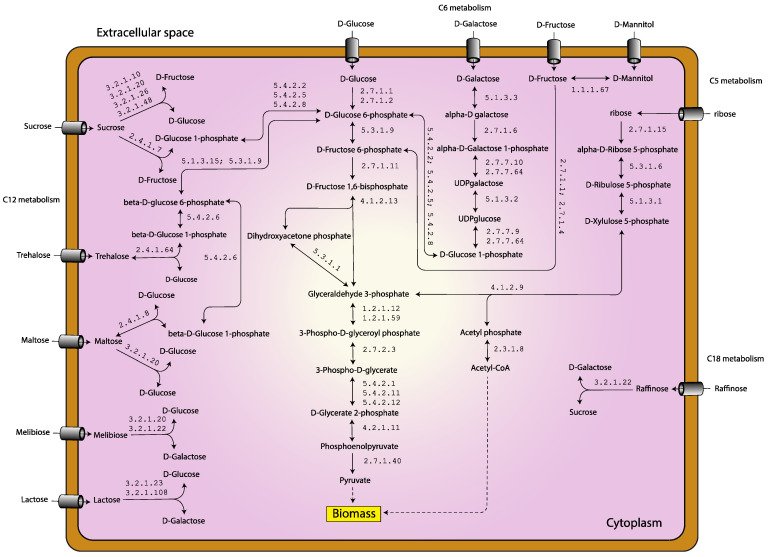
A highlight metabolic landscape of *L. reuteri* KUB-AC5 for growth and biomass production.

**Figure 3 biology-11-00294-f003:**
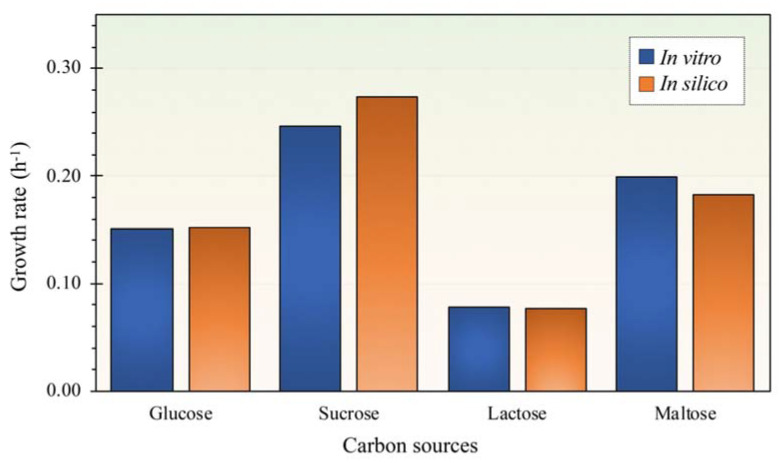
Model validation by comparison of growth rate (h^−1^) between *in silico* data and *in vitro* data across different carbon sources.

**Figure 4 biology-11-00294-f004:**
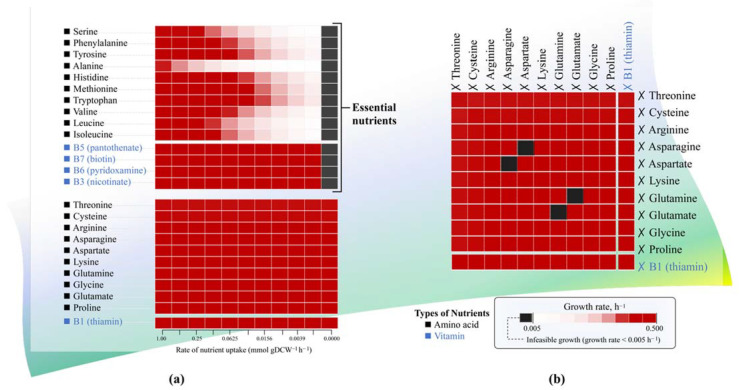
The identification of essential nutrients for *L. reuteri* KUB-AC5 growth: (**a**,**b**) The predicted growth behaviors of *L. reuteri* KUB-AC5 grown under the single omission analysis of nutrients, and under the double omission analysis of nutrients, respectively.

**Figure 5 biology-11-00294-f005:**
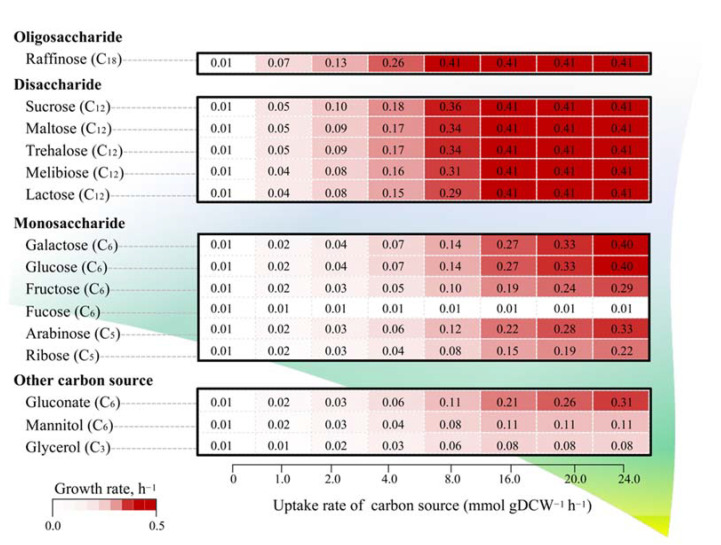
The predicted growth behaviors of *L. reuteri* KUB-AC5 grown under the different carbon sources. In addition to the color scale, the individual values contained in a matrix also represent the prediction growth rates governed under the different uptake rates of the different carbon sources.

**Figure 6 biology-11-00294-f006:**
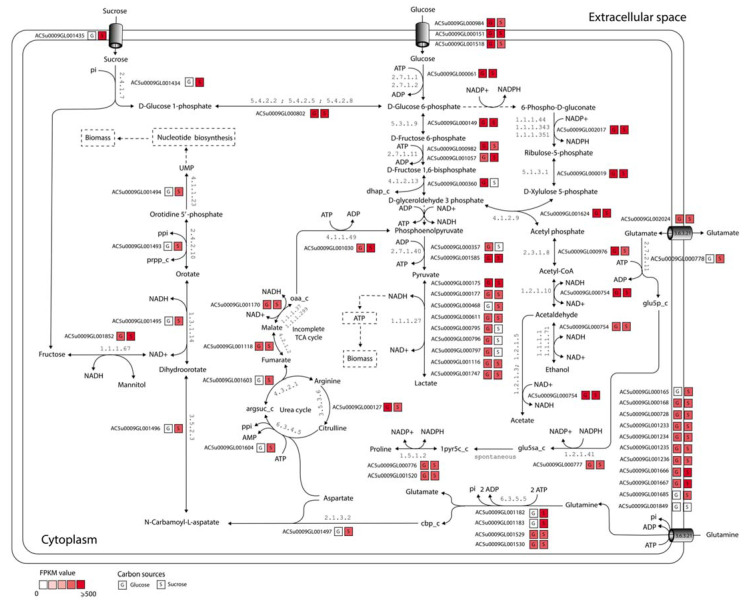
Potential metabolic routes for enhancing growth of *L. reuteri* KUB-AC5 by integrative transcriptome analysis. Abbreviated metabolite names are as follows: prpp_c, 5-phospho-alpha-ribose 1-diphosphate; argsuc_c, N(omega)-(L-Arginino)succinate; oaa_c, oxaloacetate; orn_L_c, ornithine; glu5sa_c, glutamate 5-semialdehyde; dhap_c, dihydroxyacetone phosphate; cbp_c, carbamoyl phosphate; glu5p_c, glutamate 5-phosphate; 1pyr5c_c, 1-pyrroline-5-carboxylate. Dashed lines represent multistep reactions.

**Table 1 biology-11-00294-t001:** Comparative characteristics of genomes towards metabolic networks of different strains of *L. reuteri.*

Genomic Characteristics	KUB-AC5 (This Study)	JCM 1112 ^1^	ATCC PTA 6475 ^2^
Genome size (Mb)	2.19 ^3^	2.04	2.04
Total number of protein-coding genes	2196 ^3^	1943	2019
**Metabolic Network Characteristics**	***i*TN656 (This study)**	**LbReuteri**	***i*HL622**
Total genes	656	530	622
Total metabolites	831	660	713
Total metabolic reactions	953	714	869
Gene-protein-reaction (GPR) associations	810	606	709
Compartments	2	2	2

^1^ Data was taken from [17], ^2^ from [18], and ^3^ from [13].

**Table 2 biology-11-00294-t002:** Growth kinetic characteristics of *L. reuteri* KUB-AC5 using different carbon sources.

Characteristic	Glucose	Maltose	Sucrose	Lactose
Growth rate (μ_max_, h^−1^) *	0.151 ± 0.004 ^c^	0.199 ± 0.009 ^b^	0.247 ± 0.003 ^a^	0.078 ± 0.005 ^d^
Biomass production (gDCW L^−1^) *	1.010 ± 0.026 ^c^	1.240 ± 0.053 ^b^	1.650 ± 0.030 ^a^	0.753 ± 0.025 ^d^
Substrate uptake rate (mmol gDCW^−1^ h^−1^)	6.634 ± 0.684 ^a^	3.310 ± 0.764 ^c^	5.033 ± 0.310 ^b^	0.940 ± 0.322 ^d^

* Experimental data were obtained from of the glucose, sucrose, and maltose cultures grown for 12 h and lactose for 18 h. The superscript letters indicate to the statistical differences between values of each characteristic on the different carbon sources which were analyzed by Duncan’s multiple range test under *p*-value < 0.05 as cutoff.

## Data Availability

Not applicable.

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
