# Peer review of "Probing Genome-Scale Model Reveals Metabolic Capability and Essential Nutrients for Growth of Probiotic Limosilactobacillus reuteri KUB-AC5"

_biology, 2022, doi:10.3390/biology11020294_

Round 1
Reviewer 1 Report
The manuscript provides the high quality genome-scale metabolic model (GSMM) of probiotic Limosilactobacillus reuteri KUB-AC5 (iTN656) for enhancing growth and biomass production. To explore the metabolic capability of this strain, the authors constructed a metabolic network and model of L. reuteri KUB-AC5 and afterwards simulated the growth physiology under different carbon sources. The main question posed is whether the proposed model can be used to correctly predict the metabolic capacity of L. reuteri KUB-AC5. To address this question the iTN656 model was validated by comparison of growth rate (h-1) between in silico data and in vivo data across different carbon sources. It was further exploited as a scaffold for growth simulation and integrative transcriptomics analysis.
The addressed issue is relevant in the field because the developed GSMM can be used as a systems biology tool for gaining insights into cell metabolism of other lactic acid bacteria. It can also provide a theoretical foundation for designing cultivation media and environments for scale-up of probiotic production.
Compared to other published material particularly other published models of L. reuteri the manuscript provides both: the manually curated model, and the model validated experimentally used along with transcriptomics data. The two previously published L. reuteri models (Saulnieret al. 2011, https://doi.org/10.1371/journal.pone.0018783) were used along with transcriptomics data but were not manually curated and were not used to quantitatively predict metabolic behaviour.
Two other L. reuteri models LbReuteri (Kristjansdottiret al. 2019, https://doi.org/10.1186/s12934-019-1229-3) and iHL622 (Luo et al. 2021, https://doi.org/10.1186/s12896-021-00702-w) were also manually curated and experimentally validated however they did not integrated transcriptomic data. The iTN656 showed the largest number of genes, metabolites, metabolic reactions and Gene-protein-reaction (GPR) associations in the model when compared to LbReuteri and iHL622 models.
The manuscript is up-to-date and clearly written.
Based on the obtained results the authors concluded that the developed GSMM offers a powerful tool for multi-level omics analysis, enabling probiotic strain optimization for biomass overproduction in industrial scale. This conclusion is consistent with presented evidence that the iTN656 model could show a metabolic capability under various carbon sources and guide potentially 14 essential single nutrients and 2 essential double nutrients for L. reuteri KUB-AC5 growth through single and double omission analysis. Promisingly, the iTN656 model was further integrated with transcriptome data suggesting putative metabolic routes as preferable paths e.g., sucrose uptake, nucleotide biosynthesis, urea cycle and glutamine transporter for L. reuteri KUB-AC5 growth.
The cited references are appropriate and adequate to related and previous work except for citation [16]
Generally, results are presented in clear tables and attractive figures, but Figure 1 (b) should be corrected because the numbers of genes for particular metabolic functional categories are hardly visible (too light color of fonts).
Specific comments below:
Lines 31-32
Correct as follows: “is important to reach cell concentration and cost reduction”
Lines 62-65
Revise English “As media formulation is the important factor for probiotic production in industry. Lactobacilli are thus fastidious microorganisms that require complex nutrient including carbohydrates, amino acids, vitamins, nucleotides as well as fatty acids”
Lines 71-73
Correct as follows: “Previously, a genome sequencing of L. reuteri KUB-AC5 showed its metabolic potential associated with probiotics properties, such as the ability to synthesize cobalamin (vitamin B12) and folate (vitamin B9)”
Line 82
Correct the name and citation “Lacticaseibacillus casei MG1363 [16]” Strain MG1363 is Lactococcus lactis subsp. cremoris, the cited work presents model of oleaginous fungus Mortierella alpina
Lines 89-91
Revise English “For enhancing growth and increasing biomass, an identifying the essential nutritional requirements by integrative GSMM of probiotic L. reuteri KUB-AC5 with transcriptomics was explored.”
Lines 92-94
Revise English “It can also serve for guiding the development to provide a theoretical foundation for designing cultivation media and environments for scale-up of probiotic production in the future.”
Lines 98-99
Revise English “The metabolic network of L. reuteri JCM1112 [17] was selected in this study since it was close proximity between their genomes”
Line 102
Correct as follows: “the best bidirectional (BBH) hits”
Line 149
Correct as follows: “centrifugation at 12,000 revolutions per minute (rpm) at 4 °C for 4 min”
Lines 222-223
Correct as follows: “iTN656 was developed from its genome sequence”
Lines 274-276
Revise English “According to physiological report [3, 4], the strain KUB-AC5 could tolerance on high bile concentrations in gastro-intestinal tract by hydrolyzed or deconjugated bile acid.”
Lines 278-280
Correct as follows: “The levan showed antimicrobial activity against pathogenic bacteria including Salmonellae, Listeriae, Pseudomonads, and certain strains of Escherichia coli, Staphylococcus aureus, B. cereus, and B. subtilis”
Line 350
Change „Lactobacilli” to „Lactobacilli”
Lines 379-380
Change „simulating” to „stimulating”
Line 383
Correct as follows: “oligosaccharides that are frequently found”
Line 402
Correct as follows: “Moreover, it contains enzymes involved in glycerol”
Reviewer 2 Report
The authors of the study entitled "Probing genome-scale model reveals metabolic capability and essential nutrients for growth of probiotic Limosilactobacillus reuteri KUB-AC5" present their study on a genome-scale metabolic model for L. reuteri. They firstly introduce the model and put in a context to other existing models. Further, they show simulations for suited carbon sources as well as essential amino acids and vitamins. The simulation data are substantiated by in vivo experiments with respect to different carbon sources. The sections are logically arranged and the contents contribute interesting aspects for more general knowledge in the area of probiotic lactic acid bacteria.
There are some minor points to address before publication:
- I would suggest a further careful English language editing (some missing words, syntax, grammar).
-
Line 82: for Lacticaseibacillus casei MG1363, [16] is given as a reference. However [16] handles Mortierella alpina. Please provide the correct reference.
- Line 177: The github link does not work. Please provide a publicly available source of the code.
- Figure 1b: the bars are missing - please provide the correct image.
- Section 3.3: are there any in vivo experiments that support the findings about the essential nutrients? If so, please provide some examples since this would further substantiate the model data.
Author Response
Reviewer 2:
English language and style
( ) Extensive editing of English language and style required
(x) Moderate English changes required
( ) English language and style are fine/minor spell check required
( ) I don't feel qualified to judge about the English language and style
Answer: Thank you very much for your view and comments. It has been carefully checked English throughout the revised manuscript.
|
Yes |
Can be improved |
Must be improved |
Not applicable |
|
|
Does the introduction provide sufficient background and include all relevant references? |
(x) |
( ) |
( ) |
( ) |
|
Is the research design appropriate? |
(x) |
( ) |
( ) |
( ) |
|
Are the methods adequately described? |
(x) |
( ) |
( ) |
( ) |
|
Are the results clearly presented? |
(x) |
( ) |
( ) |
( ) |
|
Are the conclusions supported by the results? |
(x) |
( ) |
( ) |
( ) |
Answer: Thank you for your view and comments.
Comments and Suggestions for Authors
The authors of the study entitled "Probing genome-scale model reveals metabolic capability and essential nutrients for growth of probiotic Limosilactobacillus reuteri KUB-AC5" present their study on a genome-scale metabolic model for L. reuteri. They firstly introduce the model and put in a context to other existing models. Further, they show simulations for suited carbon sources as well as essential amino acids and vitamins. The simulation data are substantiated by in vivo experiments with respect to different carbon sources. The sections are logically arranged and the contents contribute interesting aspects for more general knowledge in the area of probiotic lactic acid bacteria.
Answer: Thank you very much you for your view and valuable feedbacks.
There are some minor points to address before publication:
- I would suggest a further careful English language editing (some missing words, syntax, grammar).
Answer: Thank you very much for your view and comments. It has been carefully checked English throughout the revised manuscript.
- Line 82: for Lacticaseibacillus casei MG1363, [16] is given as a reference. However [16] handles Mortierella alpina. Please provide the correct reference.
Answer: We apologize for this mistake. We already corrected the cited reference in the revised manuscript (line 539, page 14).
- Line 177: The github link does not work. Please provide a publicly available source of the code.
Answer: Thank you very much for your careful review of our work. The github link is currently available.
- Figure 1b: the bars are missing - please provide the correct image.
Answer: We apologize for this oversight. The bars in Figure 1b are properly displayed in the revised manuscript.
- Section 3.3: are there any in vivo experiments that support the findings about the essential nutrients? If so, please provide some examples since this would further substantiate the model data.
Answer: Thank you very much for your comments. Here, we further compared the in silico growth behaviors of L. reuteri KUB-AC5 under the omission analysis of these identified 14 essential single nutrients against the literature. As observed, the in vitro omission experiments performed by Santos et al., 2009 [37], showed that omitted leucine, phenylalanine, valine, tyrosine, histidine, methionine or tryptophan from the chemically defined medium (CDM) at a time profoundly inhibited the growth of L. reuteri. This is in agreement with the simulation results of iTN656 suggesting that the model can be used for identifying the preferable nutrients for optimizing growth of L. reuteri. In addition, the metabolic capability of producing vitamin B1 (thiamin) by L. reuteri predicted by the iTN656 model was found consistent with earlier study by Saulnieret al. 2011 [38], which identified a complete pathway for thiamine biosynthesis in L. reuteri genome. These supportive information hereby suggested that the iTN656 model can facilitate the optimization of probiotic production in the fields of food and feed stuffs at industrial scale. These useful information already added in the revised manuscript (line 372-384, page 10). We already added useful references in the revised manuscript (line 585-589, page 15).
References:
[37] Santos, F.; Teusink, B.; Molenaar, D.; van Heck, M.; Wels, M.; Sieuwerts, S.; de Vos, W. M.; Hugenholtz, J. Effect of amino acid availability on vitamin B12 production in Lactobacillus reuteri. Appl Environ Microbiol 2009, 75 (12), 3930-3936.
[38] Saulnier, D. M.; Santos, F.; Roos, S.; Mistretta, T. A.; Spinler, J. K.; Molenaar, D.; Teusink, B.; Versalovic, J. Exploring metabolic pathway reconstruction and genome-wide expression profiling in Lactobacillus reuteri to define functional probiotic features. PLoS ONE 2011, 6(4): e18783.
Reviewer 3 Report
Authors reported the genome scale metabolic model (GSMM) of Limosilactobacillus reuteri KUB-AC5, namely iTN656. It was meaningful to know the genomic information of probiotic bacteria for the future various application. In addition, the materials & methods in this study looked appropriate. I have some comments and suggestions shown in below. Hope they would help your revised manuscript.
- According to the introduction, the rough information of L. reuteri KUB-AC5 was provided. As far as I know, probiotic bacteria are often discussed at strain level. Then I just want to make sure that what is the specific differences or characteristics among other strains of L. reuteri, what is the meaning of the construction of the GSMM for this strain. Like Kristjansdottir et al. (reference No. 17), it was natural a type strain of that species was selected to understand the bacterial characteristics. Please add the explanation of the different meaning, significance to use this strain, KUB-AC5 in this study.
- L138-9, Please add the reason that authors examined these four kinds of carbon sources.
- According to the results of Figure 1(b), I think this figure has been uncompleted. I could not see the number of genes in each metabolism.
- According to the results of Table 1, the results of KUB-AC5 and iTN656 should move to the head of the Table.
- Regarding the results of Figure 3, “in vivo” should be change to “in vitro” in whole manuscript.
Author Response
Reviewer 3:
English language and style
( ) Extensive editing of English language and style required
( ) Moderate English changes required
(x) English language and style are fine/minor spell check required
( ) I don't feel qualified to judge about the English language and style
Answer: Thank you very much for your view and comments. It has been carefully checked English throughout the revised manuscript.
|
Yes |
Can be improved |
Must be improved |
Not applicable |
|
|
Does the introduction provide sufficient background and include all relevant references? |
( ) |
(x) |
( ) |
( ) |
|
Is the research design appropriate? |
(x) |
( ) |
( ) |
( ) |
|
Are the methods adequately described? |
(x) |
( ) |
( ) |
( ) |
|
Are the results clearly presented? |
( ) |
(x) |
( ) |
( ) |
|
Are the conclusions supported by the results? |
( ) |
(x) |
( ) |
( ) |
Answer: Thank you for your view and comments. We already improved the background, included all relevant references and also improved the description of the introduction, results and conclusions in the revised manuscript.
Comments and Suggestions for Authors
Authors reported the genome scale metabolic model (GSMM) of Limosilactobacillus reuteri KUB-AC5, namely iTN656. It was meaningful to know the genomic information of probiotic bacteria for the future various application. In addition, the materials & methods in this study looked appropriate. I have some comments and suggestions shown in below. Hope they would help your revised manuscript.
Answer: Thank you very much, we highly appreciate the reviewers’ insightful and helpful comments for the manuscript.
- According to the introduction, the rough information ofL. reuteri KUB-AC5 was provided. As far as I know, probiotic bacteria are often discussed at strain level. Then I just want to make sure that what is the specific differences or characteristics among other strains of L. reuteri, what is the meaning of the construction of the GSMM for this strain. Like Kristjansdottir et al. (reference No. 17), it was natural a type strain of that species was selected to understand the bacterial characteristics. Please add the explanation of the different meaning, significance to use this strain, KUB-AC5 in this study.
Answer: Thank you very much. The strain KUB-AC5 was originally isolated from chicken intestine displayed the hallmark features of probiotics properties (Nitisinprasert et al. 2000). This strain is able to produce antimicrobial substances which inhibited both of gram-positive and gram-negative bacteria, such as Escherichia coli, Salmonella Typhimurium, Salmonella Enteritidis, Bacillus cereus, Bacillus subtilis, Micrococcus luteus, and Staphylococcus aureus (Nitisinprasert et al. 2000; Nakphaichit et al. 2011). According to in vivo assay, strain KUB-AC5 could inhibit Salmonella infection in chicken and mice model (Nakphaichit et al. 2019; Buddhasiri et al. 2021). In addition, this strain additionally attenuated inflammation in both gut and spleen of mice model (Buddhasiri et al. 2021). We already added this useful information in the Introduction in the revised manuscript (line 50-57, Page 2).
- L138-9, Please add the reason that authors examined these four kinds of carbon sources.
Answer: Thank you very much for comments. In this study, glucose, sucrose, lactose and maltose were selected for GSMM validation regarding to their implications in food and feed industries. Besides, we would like to use C6 (e.g., glucose) and C12 (e.g., sucrose, lactose and maltose) for overall metabolic capability studies. Typically, glucose is a basal carbon source used in MRS medium which is a selective culture medium designed to favor the growth of Lactobacilli in the laboratory. For lower production costs, sucrose, which is the most abundant disaccharide, has been attracting much attention as a promising carbon source for large scale production of probiotics. In addition, lactose and maltose are the two common food disaccharides, frequently derived from milks (e.g. human milk oligosaccharides (HMOs)) and starch, respectively. We already included useful information in the revised manuscript (line 286-295, page 7-8).
- According to the results of Figure 1(b), I think this figure has been uncompleted. I could not see the number of genes in each metabolism.
Answer: We apologize for this mistakes. The numbers of genes in each metabolism in Figure 1b are clearly displayed in the revised manuscript.
- According to the results of Table 1, the results of KUB-AC5 and iTN656 should move to the head of the Table.
Answer: We already moved the results of KUB-AC5 and iTN656 to the head of the Table 1 in the revised manuscript (page 6).
- Regarding the results of Figure 3, “in vivo” should be change to “in vitro” in whole manuscript.
Answer: We already edited the Figure 3 and the whole manuscript according to your comments.